# Synthesis, Structures, and Water Adsorption of Two Coordination Polymers Constructed by M(II) (M = Ni (1) and Zn (2)) with 1,3-Bis(4-Pyridyl)Propane (bpp) and 1,2,4,5-Benzenetetracarboxylate (BT^4−^) Ligands

**DOI:** 10.3390/polym12102222

**Published:** 2020-09-27

**Authors:** Chih-Chieh Wang, Wei-Cheng Yi, Zi-Ling Huang, Tsai-Wen Chang, Wen-Chi Chien, Yueh-Yi Tseng, Bo-Hao Chen, Yu-Chun Chuang, Gene-Hsiang Lee

**Affiliations:** 1Department of Chemistry, Soochow University, Taipei 11102, Taiwan; 06333004@gm.scu.edu.tw (W.-C.Y.); wendyhung51@gmail.com (Z.-L.H.); meadow8692@gmail.com (T.-W.C.); momo2013124@gmail.com (W.-C.C.); lunar24575559@gmail.com (Y.-Y.T.); 2National Synchrotron Radiation Research Center, Hsinchu 30076, Taiwan; chen.bh@nsrrc.org.tw; 3Instrumentation Center, National Taiwan University, Taipei 10617, Taiwan; ghlee@ntu.edu.tw

**Keywords:** coordination polymer, metal-organic framework, hydrogen bonds, water adsorption

## Abstract

Two coordination polymers (CPs), with chemical formulas {[Ni_2_(bpp)_2_(BT)(H_2_O)_6_] 1.5(EtOH) 1.5H_2_O}_n_ (**1**) and [Zn(bpp)(BT)_0.5_]·5H_2_O (**2**) (bpp = 1,3-bis(4-pyridyl)propane, and BT^4−^ = tetraanion of 1,2,4,5-Benzenetetracarboxylic acid), have been synthesized and structurally characterized by single-crystal X-ray diffraction methods. In compound **1**, the coordination environments of two crystallographically independent Ni(II) ions are both distorted octahedral bonded to two nitrogen donors from two bpp ligands and four oxygen donors from one BT^4-^ ligand and three water molecules. Both bpp and BT^4−^ act as bridging ligands with *bis*-monodentate and 1,4-*bis*-monodentate coordination modes, respectively, connecting the Ni(II) ions to form a 2D layered metal-organic framework (MOF). Adjacent 2D layers are then arranged orderly in an ABAB manner to complete their 3D supramolecular architecture. In **2**, the coordination environment of Zn(II) ion is distorted tetrahedral bonded to two nitrogen donors from two bpp ligands and two oxygen donors from two BT^4−^ ligands. Both bpp and BT^4-^ act as bridging ligands with *bis*-monodentate and 1,2,4,5-*tetrakis*-monodentate coordination modes, respectively, connecting the Zn(II) ions to form a 3D MOF. The reversible water de-/adsorption behavior of **1** between dehydrated and rehydrated forms has been verified by cyclic Thermogravimetric (TG) analyses through de-/rehydration processes. Compound **1** also exhibits significant water vapor hysteresis isotherms.

## 1. Introduction

Coordination compounds with open two-dimensional (2D) or three-dimensional (3D) polymeric metal-organic frameworks (MOFs) [1,2] containing guest solvent molecules are attractive research fields, owing not only to their designable structure with unusual flexibilities but also to their tunable functional application [3,4,5,6,7,8], which provide the possibility for developing advanced functional materials for assemblies of guest molecules in confined spaces [9,10]. In particular, 1,2,4,5-benzenetetracarboxylic acid (H_4_BT) and its de-protonated form of BT^4−^ appear attractive as building blocks with template capacity for the construction of 1D, 2D and 3D extended coordination polymers (CPs) [11,12,13,14,15,16,17,18,19,20,21,22,23,24,25,26,27]. In the relevant approach, BT^4−^, which possesses a rigid aromatic multi-carboxylate structure, has been used as a poly-functional ligand, including as a bridging ligand with various coordination modes to build up many CPs or MOFs with novel extended networks and also as a hydrogen bonding acceptor in the stabilization of extended 3D supramolecular networks [11,12,13,14,15,16,17,18,19,20,21,22,23,24,25,26,27,28,29]. The combination of BT^4−^ and additional rigid or flexible N-based co-ligands in such synthetic systems can generate interesting polymeric frameworks with different structural topologies [19,20,21,30,31,32,33,34,35,36]. In this regard, bi-pyridyl-type ligand, 1,3-bis(4-pyridyl)propane (bpp), is a flexible bifunctional ligand [31] that can adopt different conformations (TT, TG, GG, GG’; T = trans and G = gauche, as shown in Scheme 1) via the rotation of aliphatic chain [−CH_2_−CH_2_−CH_2_−] between two pyridyl rings to obtain different structural topologies in polymeric structures [34]. Several crystal structures containing transition metal ions and bpp ligands show the formation of 1D, 2D and 3D networks [30,31,32,33,34,35,36]. However, the polymeric structures constructed by transition metal ions with BT^4−^ and bpp ligands are rare and only a few polymeric networks have been reported in the literature [19,20,21]. Focusing on this approach, we report here the synthesis and structural characterization of two CPs, {[Ni_2_(bpp)_2_(BT)(H_2_O)_6_] 1.5(EtOH) 1.5H_2_O}_n_ (**1**) and [Zn(bpp)(BT)_0.5_]·5H_2_O (**2**), in which the BT^4−^ ligands act as the bridging ligands with two kinds of coordination modes, 1,4-*bis*-monodentate in **1** and 1,2,4,5-tetrakis-monodentate in **2** (shown in Scheme 2), and the bpp acts as a bridging ligand with *bis*-monodentate coordination mode, connecting the M(II) ions to generate 2D layered and 3D MOFs, respectively. The thermal stability, reversible de-/adsorption of the guest water molecules and water vapor ad-/desorption isotherms are the focus of this study.

## 2. Experimental Details

### 2.1. General Procedures

All chemicals were used as commercially obtained. Elemental analyses were conducted with a Perkin-Elmer 2400 elemental analyzer. IR spectra (KBr disk) were recorded on a Nicolet FT IR, MAGNA-IR 500 spectrometer. Thermogravimetric analysis (TGA) was performed on a Perkin-Elmer 7 Series/UNIX TGA7 analyzer. Single-phased powder samples were loaded into alumina pans and heated with a ramp rate of 5 °C/min from room temperature to 700 °C under a nitrogen atmosphere. The water vapor adsorption isotherm at 298 K was measured in the gaseous state by using BELSORP-max volumetric adsorption equipment from BEL, Osaka, Japan. The adsorbent sample (~100–150 mg), which was prepared at 150 °C and 10^−2^ Pa for around 24 h, was placed into the sample cell, and then the change in pressure was monitored and the degree of adsorption was determined by the decrease in pressure at equilibrium state.

### 2.2. Synthesis of {[Ni_2_(bpp)_2_(BT)(H_2_O)_6_] 1.5(EtOH) 1.5H_2_O}_n_
*(**1**)*

An ethanol/H_2_O solution (1:1, 3 mL) of 1,2,4,5-benzene-tetracarboxylic acid (H_4_BT) (0.02 mmol) was added to an ethanol/water (1:1, 6 mL) solution of NiCl_2_ (0.04 mmol) and 1,3-bis(4-pyridyl)propane (0.04 mmol) at RT. After standing for one week, blue block crystals of **1** (yield, 53.53%) were obtained. The resulting crystals were filtrated and then washed several times with distilled water. Anal. calc. for (**1**), C_3__9_H_54_N_4_O_17_Ni_2_: C 48.37, N 5.78, H 5.58; found: C 48.91, N 5.77, H 5.69. IR: ν = 3417 (vs, br), 1620 (vs), 1571 (vs), 1433 (s), 1414 (s), 1375 (vs), 1324 (m), 1222 (m), 1137 (w), 1072 (m), 1031 (s), 823 (s), 620 (m), 518 (m) cm^−1^.

### 2.3. Synthesis of [Zn(bpp)(BT)_0.5_]·5H_2_O *(**2**)*

An ethanol/H_2_O solution (1:1, 3 mL) of 1,2,4,5-benzene-tetracarboxylic acid (H_4_BT) (0.0050g, 0.02 mmol) was added to an ethanol/water (1:1, 6 mL) solution of ZnF_2_ (0.0041g, 0.04 mmol) and 1,3-bis(4-pyridyl)propane (0.0079 g, 0.04 mmol) at RT. After standing for several days, colorless needle-like crystals of **2** (yield, 55.8%) were obtained. The resulting crystals were filtrated and then washed several times with distilled water. Anal. calc. for (**2**), C_18_H_25_N_2_O_9_Zn: C 45.15, N 5.84, H 5.22; found: C 45.14, N 5.85, H 5.34. IR: ν = 3358 (vs, br), 1620 (s), 1571 (s), 1484 (m), 1431 (s), 1378 (vs), 1332 (m), 1226 (w), 1137 (w), 1071 (w), 922 (w), 839 (m), 805 (m),683 (m), 666 (m), 585 (m), 531 (m) cm^−1^.

### 2.4. X-ray Crystallography and Refinements

The diffraction data for compounds **1** and **2** were collected on a Siemens SMART diffractomer with Mo radiation (λ = 0.71073 Å) at 150 K. Cell parameters were retrieved using SMART [37] software and refined with SAINT [38] on all observed reflections. Data reduction was performed with the SAINT [39] software and corrected for Lorentz and polarization effects. Absorption corrections were applied with the program SADABS [39]. Direct phase determination and subsequent difference Fourier map synthesis yielded the positions of all atoms. The final full-matrix, least-squares refinement on *F*^2^ was applied for all observed reflections [I > 2σ(I)]. All calculations were performed by using the SHELXTL-PC V 5.03 software package [40]. Crystal data and details of the data collection and structure refinements for **1** and **2** are summarized in Table 1. CCDC-2022795 and 2,022,797 for **1** and **2**, respectively, contain the supplementary crystallographic data for this paper (Appendix A). These data can be obtained free of charge from www.ccdc.cam.ac.uk/conts/retrieving.html or from the Cambridge Crystallographic Data Centre, 12, Union Road, Cambridge CB2 1EZ, UK; fax: (internat.) +44-1223/336-033; email: deposit@ccdc.cam.ac.uk.

### 2.5. In Situ X-ray Powder Diffraction

The powder X-ray diffraction measurement of compounds **1** and **2** was performed at the BL01C2 beamline of the National Synchrotron Radiation Research Center (NSRRC) in Taiwan. The X-ray energy was selected at 1.0332 Å. The diffraction pattern was recorded with a Mar345 imaging-plate detector and typical exposure duration of 2 min. The one-dimensional powder diffraction profile was converted with GSAS-II program [41]. The diffraction angles were calibrated according to Bragg positions of lanthanum boride (SRM660b) standards.

## 3. Results and Discussion

### 3.1. Synthesis and IR Spectroscopy

Compounds **1** and **2** were synthesized by direct mixing of M(II) salts, bpp and H_4_BT with a molar ratio of 2:2:1 to obtain blue rod-like crystals of **1** and colorless needle-like crystals of **2**, respectively. The synthetic representation is shown in Scheme 3. The most relevant IR features are those associated with the BT^4−^ and bpp ligands [19,20]. Absorption bands in the range of 1400–1700 cm^−1^ can be related to the carboxylate groups of BT^4−^ ligand and bpp ligand, since the main vibrational frequencies of the two ligands appear in the same spectral region. However, a strong band centered at around 1571 cm^−1^ in the spectrum can be attributed to vibrational modes representing the C–C and C–N mixing stretching motions and is in agreement with the characteristics of the bpp ligand. Additional broad bands appear in the range of 3100–3500 cm^−1^ in **1** and **2**, indicating the presence of the O–H stretching vibration from water molecules.

### 3.2. Structural Characterization of {[Ni_2_(bpp)_2_(BT)(H_2_O)_6_] 1.5(EtOH)·1.5H_2_O}_n_
*(**1**)*

Compound **1** is iso-structural with that of {[Co_2_(bpp)_2_(BT)(H_2_O)_6_]·2H_2_O}_n_ [20] and crystallizes in the monoclinic *C*c space group, in which the asymmetric unit is composed of two Ni(II) centers, two bpp, one BT^4−^ ligand, six coordinated H_2_O molecules and one and a half solvated H_2_O and C_2_H_5_OH molecules. The molecular structures of **1** are shown in Figure 1a, in which two crystallographically independent Ni(II) ions are both six coordinate bonded to four oxygen donors of one BT^4−^ ligand and three H_2_O molecules and two nitrogen donors of two bpp ligands located at the *cis* position, forming a distorted octahedral geometry. Bond lengths and angles around the Ni(II) ions are listed in Appendix A (included in the Appendix A). In **1**, two crystallographically independent bpp ligands, adopting TT and TG conformations, respectively, both act as bridging ligands with *bis*-monodentate coordination mode connecting the Ni(II) ions to form one-dimensional (1D) zigzag chains (Figure 1b). Adjacent chains are inter-linked via the connectivity between Ni(II) ions and BT^4−^ ligands with 1,4-*bis*-monodentate coordination mode [20] to generate a 2D layered MOF (Figure 1b). The Ni(II)⋅⋅⋅Ni(II) separations via TT-, TG-bpp and BT^4−^ bridges are 12.922(3), 12.203(3) and 11.220(1) Å, respectively. Adjacent 2D layers are then arranged orderly in an ABAB parallel manner to build up their 3D supramolecular architectures (Figure 1c). It is important to note that intra- and inter-layer O–H⋅⋅⋅O hydrogen bonding interactions between the coordinated H_2_O molecules and uncoordinated oxygen atoms of BT^4−^ ligands, with O⋅⋅⋅O distances in the range of 2.675(1)–3.170(2) Å, provide extra stabilization energies for the construction of its 3D structures. Furthermore, guest solvent molecules, 1.5 EtOH and 1.5 H_2_O, which are located at the vacant spaces in the 3D supramolecular architecture, are reinforced by intermolecular hydrogen bonding interactions between O–H groups of guest molecules and oxygen atoms of coordinated H_2_O molecules and BT^4−^ ligands. Relevant parameters of hydrogen bonds are summarized in Appendix A (included in the Appendix A).

The thermal stability and temperature-dependent structural variation for **1** were studied by thermogravimetric (TG) analysis and in situ powder X-ray diffraction (PXRD) measurements, respectively. During the heating process, the TG analysis of **1** displayed a two-step weight loss (Figure 2a), while the first weight loss of 20.3% corresponded to the weight losses of six coordinated water molecules, 1.5 guest ethanol and 1.5 water molecules (calc. 21.1%), occurring in the range of approximately 33.9–156.9 °C and then remaining thermally stable up to 198.4 without any weight loss. On further heating, these samples decomposed. It is important to note that, during the heating processes, crystals in **1** were accompanied by morphological changes. The simultaneous and gradual color changes of the crystals were followed by optical microscopy, as shown in Figure 2a, during the de-solvation processes at some specific temperatures. A well-formed blue crystal of **1** was obtained. As the temperature was raised, the colors of dried crystals gradually turned from blue to pale-blue, and many chaps with random cracks on the crystal surface were observed. In order to gain more structural information, in situ temperature-dependent synchrotron PXRD patterns of **1** were collected from 25 °C to 410 °C, and the results at some specific temperatures are shown in Figure 2b. The PXRD patterns of as-synthesized samples **1** at RT matched well with the simulated one based on the single crystal structures. As the temperature increased, solvent de-solvation processes were initialized at 100 °C for **1** and converted to a meta-stable phase before entire structure collapse. However, due to the poor long-range ordering of PXRD data, the unit cells of dehydrated form **1** could not be determined directly by using synchrotron PXRD data. The temperature-dependent PXRD results are consistent with the TGA results and morphological changes.

### 3.3. Structural Description of {[Zn(bpp)(BT)_0.5_]·5H_2_O}_n_
*(**2**)*

Compound **2** crystallizes in the monoclinic *P* 2_1_/c space group, in which the asymmetric unit is composed of one Zn(II) ion, one bpp, half of a BT^4−^ ligand and five guest water molecules. The coordination geometry of Zn(II) ion in **2** is distorted tetrahedral (Figure 3a) bonded to two oxygen donors from two BT^4−^ ligands and two nitrogen donors from two bpp ligands. Relevant bond lengths and angles around the Zn(II) ion are listed in Appendix A (included in the Appendix A). In **2**, BT^4−^ acts as a bridging ligand with 1,2,4,5-*tetrakis*-monodentate coordination mode [28,29] connecting the Zn(II) ions to form a 2D sinusoidal-like [Zn_2_(BT)]_n_ layered framework (Figure 3b). Adjacent layers are inter-linked via the connectivity between Zn(II) ions and bpp ligands with *bis*-monodentate coordination mode to complete its 3D MOF (Figure 3c), in which five guest water molecules are located at the vacant sites (Figure 3d, right). The Zn(II)⋅⋅⋅Zn(II) separation via the TG-bpp bridge is 10.681(1) Å and via BT^4−^ bridges is 6.170(1), 9.208(1), 11.111(3), 11.057(2) Å (Figure 3c, right), respectively. It is noteworthy that five guest water molecules are assembled together to generate a 2D sinusoidal-like hydrogen-bonded water layer (Figure 3d, left and middle), which is built up by two water cluster units—one is a (H_2_O)_6_ chair-like water ring and the other is a (H_2_O)_18_ water ring (Figure 3d, left)—via the intermolecular O–H⋅⋅⋅O hydrogen bonding interactions with the O⋅⋅⋅O distances in the range of 2.725(1)–2.818(1) Å. Importantly, synergistic O−H⋅⋅⋅O hydrogen bonding interactions between guest H_2_O molecules in the 2D water layer and carboxylate groups of BT^4−^ ligands in two [Zn_2_(BT)]_n_ layers (Figure 3d, right) provide significant stabilization energy to maintain the accumulation of 2D water layers in the 3D MOF. Relevant parameters for hydrogen bonds are summarized in Appendix A (included in Appendix A).

During the heating process, the TG analysis (see Figure 4a) revealed that **2** underwent a two-step weight loss, while the first weight loss of 18.1% corresponded to the loss of five guest H_2_O molecules (calc. 18.8%), occurring in the range of approximately 30.6–64.3 °C, and then remaining thermally stable up to 291.7 °C without any weight loss. On further heating, the sample decomposed. It is important to note that, during the heating processes, crystal **2** was accompanied by morphological changes. The simultaneous and gradual changes of the crystals were followed by optical microscopy, as shown in Figure 4a, during the dehydration process at some specific temperatures. A well-formed colorless crystal of **2** was obtained. As the temperature was raised, the color of the crystals gradually turned from colorless to white and many chaps with random cracks on the crystal surface were observed. To gain more information on the temperature-dependent structural variation, in situ synchrotron PXRD patterns of **2** were collected from 25 °C to 410 °C, and the results at some specific temperatures are shown in Figure 4b. The PXRD patterns of as-synthesized sample **2** matched well with the simulated pattern based on the single crystal structure. As the temperature increased, a new phase was transformed at 110 °C and could be sustained up to 320 °C. According to the cell refinement results, the cell volume reduced from 2078.88 Å^3^ to 1847.39 Å^3^, roughly a 12% reduction. The symmetry was kept in monoclinic lattice, and corresponding cell parameters of hydrated and dehydrated forms were *a* = 12.5099 Å, *b* = 14.7892 Å, c = 10.8858 Å, β = 113.469° and *a* = 10.6860 Å, *b* = 16.5976 Å, c = 12.4611 Å, β = 109.845°, respectively. The temperature-dependent PXRD results are consistent with the TGA results and morphological changes.

### 3.4. Water De-/Adsorption Behaviors of CPs ***1*** and ***2*** by Cyclic TG Analysis

In order to certify the reversibility of water de-/adsorption behaviors in **1** and **2** under water vapor, cyclic TG measurements were performed under de-/rehydration procedures by heating/cooling processes. Compound **1** showed a 20.0% weight decrease, corresponding to the losses of six coordinated H_2_O molecules, 1.5 guest EtOH and 1.5 H_2_O molecules, to obtain a dehydrated form of **1** after heating up to 120 °C (Figure 5a). When the dehydrated samples **1** were exposed to the water vapor and gradually cooled down to RT, the dehydrated samples showed a weight-increasing of 9.3%, approximately equal to 5.0 H_2_O molecules, to form the rehydration ones. Such heating and cooling processes were repeated for five times (Figure 5a), showing almost equally weighted increasing/decreasing percentages in the range of 9.3~10.1%, to prove the stable reversibility of the water de-/rehydration behavior. On the contrary, compound **2** shows irreversible water de-/adsorption behavior during the cyclic TG measurements shown in Figure 5b, in which a 20.0% weight decrease was found in the first cycle, corresponding to the losses of five guest H_2_O molecules, to obtain the dehydrated form **2** after heating up to 150 °C (Figure 5b). When the dehydrated samples **2** were exposed to the water vapor and cooled down to RT, the water molecules could not be re-adsorbed. The results described above show reversible water de-/adsorption behaviors between dehydrated and rehydrated forms, but **2** shows irreversible water de-/adsorption behavior. The poor water adsorption ability of dehydrated **2** may be attributed to the structural change of the dehydrated form, which may prevent the re-adsorption of water molecules via the synergistic hydrogen bonding interactions between water molecules and [Zn_2_(BT)]_n_ layers.

### 3.5. Water Sorption Studies of CP ***1***

According to cyclic TG analyses, compound **1** shows the reversible de-/adsorption behavior of approximately five water molecules. In order to explore the water adsorption ability of **1**, water vapor sorption isotherms of pre-treated dehydrated form **1** were measured at 298K. The isotherm curve (Figure 6) of dehydrated **1** shows a steady increase in adsorbed water vapor at 0 < relative P/P_0_ < 0.89, with maximum value of 210.7 cm^3^ g^−1^ at relative P/P_0_ equal to 0.89, approximately equal to 8.20 H_2_O molecules, indicating six coordinated and two guest H_2_O molecules in **1** being re-adsorbed. It is worth noting that the desorption curve did not trace the adsorption curve, exhibiting a significant hysteresis loop with a value of 148.5 cm^3^ g^−1^, approximately equal to 5.8 H_2_O molecules at lower relative P/P_0_ equal to 0.22. The water ad-/desorption isotherms of dehydrated form **1** reveal that the vacant sites of Ni(II) ions provide the opportunity for bond re-formation with water molecules as the dehydrated samples being exposed to water vapor. Recently, MOFs with high porosity have been examined for their water capture properties and found to be highly promising materials [42,43,44]. However, water capture properties applied to 3D supramolecular architectures assembled via CPs or MOFs are seldom detected and only a few cases have been investigated [45,46,47]. The water sorption isotherm with large hysteresis loops found in **1** is interesting and may be exploited as a water harvesting material.

## 4. Conclusions

Two coordination polymeric frameworks, {[Ni_2_(bpp)_2_(BT)(H_2_O)_6_] 1.5(EtOH)·1.5H_2_O}_n_ (**1**) and [Zn(bpp)(BT)_0.5_]·5H_2_O (**2**), have been structurally investigated. In compound **1**, the 3D supramolecular architecture is built up via the assembly of 2D MOFs, which are constructed through the bridges of Ni(II) with 1,4-*bis*-monodentate bpp and BT^4−^ ligands. Compound **2** is a 3D MOF constructed through the bridges of Zn(II) with *bis*-monodentate bpp and 1,2,4,5-*tetrakis*-monodentate BT^4−^ ligands. Interestingly, the guest water molecules in **2** are assembled together to generate a 2D sinusoidal-like hydrogen-bonded water layer, which is built up by two water cluster units—one is a (H_2_O)_6_ chair-like water ring and the other is a (H_2_O)_18_ water ring—via the intermolecular O–H⋅⋅⋅O hydrogen bonding interactions. Notably, compound **1** exhibits interesting water hysteresis isotherms and undergoes reversible water de-/adsorption behavior between the dehydrated and rehydrated species during thermal re-/dehydration processes. Water capture uptake observed in **1** displayed significant hysteresis loops in water ad-/desorption isotherms, which may be developed for potential application as a water harvesting material.

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
