# Peer review of "Synthesis, Structures, and Water Adsorption of Two Coordination Polymers Constructed by M(II) (M = Ni (1) and Zn (2)) with 1,3-Bis(4-Pyridyl)Propane (bpp) and 1,2,4,5-Benzenetetracarboxylate (BT4−) Ligands"

_polymers, 2020, doi:10.3390/polym12102222_

Round 1

Reviewer 1 Report

The Ms submitted by Dr. Wang reports the synthesis of two coordination polymers, in which Ni(II) and Zn(II) ions are bridging simultaneously to 1,3-bis(4-pyridyl)propane (bpp) and 1,2,4,5-benzenetetracarboxylate (BT4-), with bis-monodentate and 1,2,4,5-tetrakis-monodentate coordination modes, respectively.  The Ni(II) complex (1) forms 2D MOF, whereas  the Zn(II) complex (2) displays a 3D MOF structure. The two complexes were structurally characterized by single crystal X-ray crystallography. The reversible water de-/ad-sorption behavior of 1 between de-hydrated and re-hydrated forms has been verified by cyclic TG analyses through de-/re-hydration processes. The results are presented and discussed in a reasonable way.  However, some points and suggestions should be considered before accepting the Ms:

1)      Some references which are directly related to the 1,2,4,5-benzenetetracarboxylate (BT4-) bridging modes, as those reported in this Ms were ignored and it should be cited.  Please see S.S. Massoud et al., Inorganica Chimica Acta 370 (2011) 435; F.A. Mautner et al., Polyhedron, Polyhedron 54 (2013) 158.

2)      The English of the Ms, as I noticed some typos and grammatically errors.  Here are some examples:

Last line of abstract: Compound 1 also exhibit (exhibits)

Section 2.3: filtrated and thyen (then)

Paragraph after Fig. 1: became from blue to pale-blue, (change to: turns from blue to pale-blue or changes from …..)

In section 3.4: re-adsorbed to become re-hydrated sample 1 (needs to reward); to the losses of 5 guest H2O (loss of 5 H2O molecules of crystallization, not sure if “guest is appropriate in this case); These results described above evidence that 1 shows reversible (The results described above show reversible ….)

Conclusions: In compound 1, it 3D supramolecular (…, the 3D ….); , compound 1 exhibit (exhibits); Water capture uptakes observed in 1 display the (displays or displayed ….)

3)      The expression of “re-adsorbed to become re-hydrated” does not seem right because adsorption is different from hydration and I would suggest using “hydration and dehydration”.  The solvent H2O is either coordinated in 1 or water of crystallization in 2.

4)      Section 3.3 for compound 2, looks like similar in wording to the discussion presented for compound 1. This section should be revised carefully

Author Response

  1. Some references which are directly related to the 1,2,4,5-benzenetetracarboxylate (BT4-) bridging modes, as those reported in this Ms were ignored and it should be cited.  Please see S.S. Massoud et al., Inorganica Chimica Acta 370 (2011) 435; F.A. Mautner et al., Polyhedron, 54 (2013) 158.

Respond: Thanks for reviewer’s suggestion. These two papers have been added as ref 30 and ref 31 in the references parts and cited in the revised manuscript.

  1. The English of the Ms, as I noticed some typos and grammatically errors.  Here are some examples:

Last line of abstract: Compound 1 also exhibit (exhibits)

Section 2.3: filtrated and thyen (then)

Paragraph after Fig. 1: became from blue to pale-blue, (change to: turns from blue to pale-blue or changes from …..)

In section 3.4: re-adsorbed to become re-hydrated sample 1 (needs to reward); to the losses of 5 guest H2O (loss of 5 H2O molecules of crystallization, not sure if “guest is appropriate in this case); These results described above evidence that 1 shows reversible (The results described above show reversible ….)

Conclusions: In compound 1, it 3D supramolecular (…, the 3D ….); , compound 1 exhibit (exhibits); Water capture uptakes observed in 1 display the (displays or displayed ….)

Respond: Thanks for reviewer’s suggestions. All the mistakes of typos and grammatically errors have been corrected and the revised manuscript has been re-checked carefully.

  1. The expression of “re-adsorbed to become re-hydrated” does not seem right because adsorption is different from hydration and I would suggest using “hydration and dehydration”. The solvent H2O is either coordinated in 1 or water of crystallization in 2.

Respond: Thanks for reviewer’s suggestion. About the expression of “re-adsorbed to become re-hydrated”, the statement has been changed to “the dehydrated samples showed a weight-increasing of 9.3 %, approximately equal to 5.0 H2O molecules, to form the re-hydration ones.”

  1. Section 3.3 for compound 2, looks like similar in wording to the discussion presented for compound 1. This section should be revised carefully.

Respond: Thanks for reviewer’s suggestion. Section 3.3 has been checked carefully and some typos and grammatically errors have been corrected in the revised manuscript.

Reviewer 2 Report

The authors describe the synthesis and characterization of two coordination polymers obtained using nickel or zinc as metals, and a bipyridine structure and a tetraacid as ligands. The latter proves capable of coordinating the metal by exploiting 2 acid groups (in the case of nickel) or 4 acid groups (in the case of zinc). The structure was determined at the atomic level by X-ray diffraction and this should guarantee ample structural certainty. Furthermore, the authors show that these coordination polymer cells sussefully trap water molecules.
The work is well done, the characterization of the metarials is done correctly, and the manuscript is easy to read, so I don't find any problems for publication.
However, some doubts remain on the application of these materials which at the moment seem to me a mere characterization and therefore risk finding a very limited audience of readers.

Author Response

The authors describe the synthesis and characterization of two coordination polymers obtained using nickel or zinc as metals, and a bipyridine structure and a tetraacid as ligands. The latter proves capable of coordinating the metal by exploiting 2 acid groups (in the case of nickel) or 4 acid groups (in the case of zinc). The structure was determined at the atomic level by X-ray diffraction and this should guarantee ample structural certainty. Furthermore, the authors show that these coordination polymer cells successfully trap water molecules.
The work is well done, the characterization of the materials is done correctly, and the manuscript is easy to read, so I don't find any problems for publication.
However, some doubts remain on the application of these materials which at the moment seem to me a mere characterization and therefore risk finding a very limited audience of readers.

Respond: Thanks for reviewer’s comments.

Reviewer 3 Report

The manuscript is devoted to synthesis, structures, and water adsorption of two new coordination polymers. This is an actual issue because polymeric metal-organic frameworks containing guest solvent molecules are attractive as functional materials. The work was performed at a fairly high experimental level. The coordination polymers used are well characterized, and the data obtained by the authors on their reversible de-/ad-sorption of the guest water molecules are of practical importance. In general, the article may be of interest to those chemists who are engaged in new materials development and, therefore, may be published in the Polymers.

Remarks:

  1. In my opinion, the manuscript would greatly improve the data on the effect of other substances (for example, methanol or dioxane) on water adsorption. If the synthesized coordination polymers are capable of adsorbing water sufficiently selectively, this opens up broad prospects for their practical use as desiccants.
  2. Here is the only typo I noticed:

Line 271. It should probably be written "Water sorption studies of CP 1" instead of “Water sorption studies of CP 2.”

Author Response

  1. In my opinion, the manuscript would greatly improve the data on the effect of other substances (for example, methanol or dioxane) on water adsorption. If the synthesized coordination polymers are capable of adsorbing water sufficiently selectively, this opens up broad prospects for their practical use as desiccants.

Respond: Thanks for reviewer’s opinion. Actually, the cyclic TGA of methanol and ethanol have been performed, but the adsorption ability is weak both for methanol and ethanol vapors.

  1. Here is the only typo I noticed:

Line 271. It should probably be written "Water sorption studies of CP 1" instead of “Water sorption studies of CP 2.”

Respond: Thanks for reviewer’s opinion. The mistake is corrected in the revised manuscript.

Round 2

Reviewer 1 Report

I do not have any specific comment, except the amount of work is too small and it does not add any thing new or even improve our knowledge in this area of research.